# Preclinical Evaluation of Artesunate as an Antineoplastic Agent in Ovarian Cancer Treatment

**DOI:** 10.3390/diagnostics11030395

**Published:** 2021-02-26

**Authors:** Anthony McDowell, Kristen S. Hill, Joseph Robert McCorkle, Justin Gorski, Yilin Zhang, Ameen A. Salahudeen, Fred Ueland, Jill M. Kolesar

**Affiliations:** 1Department of Obstetrics and Gynecology, Division of Gynecologic Oncology, College of Medicine, University of Kentucky, Lexington, KY 40536, USA; amcdowell@uky.edu (A.M.J.); justin.gorski@uky.edu (J.G.); fuela0@uky.edu (F.U.); 2Markey Cancer Center, University of Kentucky, Lexington, KY 40536, USA; Kristen.hill@uky.edu (K.S.H.); rob.mccorkle@uky.edu (J.R.M.); 3Tempus Labs, 600 W Chicago Ave. Ste 510, Chicago, IL 60654, USA; Yilin.zhang@tempus.com (Y.Z.); Ameen@tempus.com (A.A.S.); 4Department of Pharmacy Practice and Research, College of Pharmacy, University of Kentucky, Lexington, KY 40536, USA

**Keywords:** artesunate, ovarian cancer, dihydroartemisinin, Artemesia annua, carboplatin, paclitaxel

## Abstract

Background: Ovarian cancer is the deadliest gynecologic malignancy despite current first-line treatment with a platinum and taxane doublet. Artesunate has broad antineoplastic properties but has not been investigated in combination with carboplatin and paclitaxel for ovarian cancer treatment. Methods: Standard cell culture technique with commercially available ovarian cancer cell lines were utilized in cell viability, DNA damage, and cell cycle progression assays to qualify and quantify artesunate treatment effects. Additionally, the sequence of administering artesunate in combination with paclitaxel and carboplatin was determined. The activity of artesunate was also assessed in 3D organoid models of primary ovarian cancer and RNAseq analysis was utilized to identify genes and the associated genetic pathways that were differentially regulated in artesunate resistant organoid models compared to organoids that were sensitive to artesunate. Results: Artesunate treatment reduces cell viability in 2D and 3D ovarian cancer cell models. Clinically relevant concentrations of artesunate induce G1 arrest, but do not induce DNA damage. Pathways related to cell cycle progression, specifically G1/S transition, are upregulated in ovarian organoid models that are innately more resistant to artesunate compared to more sensitive models. Depending on the sequence of administration, the addition of artesunate to carboplatin and paclitaxel improves their effectiveness. Conclusions: Artesunate has preclinical activity in ovarian cancer that merits further investigation to treat ovarian cancer.

## 1. Introduction

The American Cancer Society estimates 21,410 new cases of ovarian cancer in the United States in 2021 [1]. Ovarian cancer accounts for only 1.2% of all new cancer cases but will result in 2.3% of all cancer deaths. With a 5-year overall survival of less than 50%, ovarian cancer is the deadliest gynecologic malignancy. Despite these grim statistics, there has been little improvement in patient outcomes since the early 2000s. Gynecologic Oncology Group (GOG) study 111 first showed improved survival with paclitaxel and cisplatin versus cyclophosphamide and cisplatin [2]. This was followed by GOG 158, which demonstrated the equivalence of carboplatin and paclitaxel compared to cisplatin and paclitaxel with decreased toxicity [3]. Since 2003, the carboplatin and paclitaxel doublet has been the standard of care for the adjuvant treatment of advanced ovarian cancer. To date, no trials demonstrate improvement in overall survival with the addition of a third cytotoxic agent. In 2004, GOG 182 investigated the standard combination of carboplatin and paclitaxel in sequential doublets or triplet regimens with gemcitabine, topotecan, or liposomal doxorubicin. The results showed no difference in progression-free survival and concluded that carboplatin and paclitaxel should remain standard of care [4]. Recently, therapeutic advances incorporating PARP and VEGF inhibitors into upfront and maintenance therapy show promising results in BRCA-mutated and other homologous recombination deficiency (HRD) cancers [5,6], but only 10–14% of women with epithelial ovarian cancer have a germline mutation in BRCA1 or 2 [7]. Furthermore, up to 80% of patients with advanced ovarian cancer who undergo a combination of platinum and taxane-based chemotherapy will develop recurrence. These outcomes demonstrate the need to identify additional therapeutic regimens that can be utilized to better treat patients with ovarian cancer.

Artesunate is synthesized from artemisinin, an extract from the sweet wormwood plant, *Artemisia annua* [8], and has been used as a fever reducer in Chinese herbal medicine for over 2000 years. Artesunate is currently a standard, initial treatment for malaria, but has also been shown to have antineoplastic activity across a broad spectrum of malignancies [9]. It is metabolized to a more active form, dihydroartemisinin (DHA), through plasma esterase and hepatic CYP3A4. Both artesunate and DHA are rapidly cleared from the body with half-lives of approximately 10 min and 1 h, respectively [10]. There are multiple proposed mechanisms of action by which artesunate eradicates *Plasmodium* species, including generation of free radicals through a heme-iron-dependent mechanism, inhibition of redox cycling, and interference with *Plasmodium* sarcoplasmic/endoplasmic calcium ATPase (SERCA) [8,11,12]. However, there is less consensus on its mechanistic role in cancer treatment [9,13,14,15,16,17,18].

Several in vitro studies investigated the effects of artesunate on a wide range of malignancies. In 2003, the NCI expanded this investigation to a panel of 55 cancer cell lines that further confirmed artesunate’s in vitro activity across various cancers. Furthermore, artesunate is unaffected by many of the typical drug resistance pathways [12]. Artesunate has been shown to have clinical activity in ovarian cancer treatment [15] but has not been studied in combination with current first-line therapies. In addition to the promising in vitro studies, artesunate was investigated for safety and tolerability as an addition to standard chemotherapy regimens, showing early efficacy signals and good tolerability [19]. These findings are reinforced by the numerous malaria trials investigating its safety and tolerability [10]. Given the significant evidence of in vitro and in vivo activity, we look to expand the existing preclinical knowledge of artesunate in the treatment of ovarian cancer [13,20,21]. This study aims to determine the activity and optimal timing of artesunate in combination with paclitaxel and carboplatin.

## 2. Materials and Methods

### 2.1. Cell Culture and Artesunate

Commercially available, human ovarian cancer cell lines UWB1.289 (ATCC CRL-2945), Caov-3 (ATCC HTB-75), and OVCAR-3 (ATCC HTB-161) were obtained from ATCC. Cell lines were cultured and maintained in cell line-specific, complete growth media as recommended by ATCC. All cells were incubated at 37°C in 5% CO₂. Artesunate was purchased from MedChem Express, dissolved in DMSO as a 200 mM stock solution, and stored at −80°C. Artesunate was serially diluted in DMSO, and then media, to the desired concentrations at the time of each experiment.

### 2.2. Cell Viability Assays

White-walled 96-well microplates were seeded at 3 × 10^3^ cells per well in 100 μL growth media and incubated for 24 h at 37°C, 5% CO_2_. The growth media was removed and replaced with fresh media containing serially diluted drugs or drugs of interest. We tested each drug concentration in duplicate and vehicle (0.1% DMSO) media for control, in triplicate assays. We used twelve dilutions of the artesunate stock solution, ranging from 0.0011–200 μM to treat cells and incubated them for 72 h. Cell viability was assessed using a CellTiter-Glo 2.0 viability assay (Promega) and luminescence was measured using a Varioskan LUX multimode microplate reader (ThermoFisher Scientific). We calculated percent viability by normalizing the relative luminescence signal of each treated well to the matched vehicle controls. After graphing the calculated percent viability for each artesunate concentration, a four-parameter log-logistic model was used to fit a non-linear regression line and the IC50 was calculated for each cell line using GraphPad Prism 5.01.

### 2.3. 3D Organoids

#### 2.3.1. Tumor Organoid (TO) Development

We collected ovarian tumor tissue from patients after written informed consent at the time of debulking surgery per the University of Kentucky Institutional Review Board. These samples were then dissociated and established in Matrigel^®^ Growth Factor Reduced Basement Membrane Matrix (Corning) in vitro using factor-defined media and standard growth conditions [22,23]. Representative sections of TOs were H&E stained and compared with the primary tumor. Growth and testing of the organoids were done in a commercial laboratory, Tempus Labs. Mutational concordance was performed by comparison of sequencing results between the primary tumor specimen and the resultant tumor organoids.

#### 2.3.2. Chemosensitivity Screens

Established TOs were enzymatically dissociated into single cells and plated in 384-well plates. The cells were cultured for 72 h before administering artesunate at different doses (0, 50 nM, 500 nM, 5 μM, and 50 μM). After culturing for an additional 72 h, organoids were incubated with Hoechst nuclear counterstain and imaged on a spinning disc confocal high content imager. Once imaging was completed, viability was measured by employing the MTS assay (Promega). We generated cell viability curves and IC50 values using GraphPad Prism (v5.01).

#### 2.3.3. Gene Expression Analysis

We extracted total RNA from primary ovarian cancer cell lines used in the development of the 3D organoids with RNeasy Plus Universal Mini Kit (Qiagen) followed by full- whole transcriptome sequencing. We used TruSeq Stranded Total RNA Prep Kit (Illumina) to generate libraries sequenced as 100 base pairs, single-end reads via an Illumina HiSeq platform. For all subsequent analysis, the organoid models were divided into two groups based on their sensitivity to artesunate; the sensitive group consisted of 4 models (2238, 2326, 1236, and 1267) which each had IC50s for artesunate of less than 100 nM, while the resistant group consisted of two models (1226 and 1254) which had IC50 values greater than 2 µM. The raw counts, generated from the Illumina HiSeq platform, of the samples in comparison were first normalized within samples using counts per gene per million mapped reads (CPM). We excluded from the analysis genes that were unexpressed or lowly expressed (no sample with CPM > 1). The read counts were further normalized between samples using TMM (Trimmed Means of M values) to account for the library size variance. We fit the read counts to a negative binomial distribution model to estimate variance and used the R software package, edgeR Normalization, for expression modeling and difference testing [24]. The package applies the negative binomial distribution model to detect differentially expressed genes in the RNAseq data and calculate a statistical *p*-value. The false discovery rate (FDR) was used to adjust for multiple comparisons. The unsupervised hierarchical clustering analysis and corresponding heatmap were conducted using the agglomeration method via function heatmap.2 from the R package gplots [25]. Subsequent pathway analysis for differentially expressed genes was conducted using the goana function from the R package limma [26]. This analysis, which performs gene category over-representation analysis on differentially expressed genes incorporating the effect of selection bias from transcript length, was used to identify cellular pathways which were either up or downregulated in the artesunate-resistant organoid models when compared to the sensitive models. A multiple comparison correction was not performed on the *p*-values of the identified differentially expressed pathways.

### 2.4. DNA Damage Assay

Caov-3 cells were seeded into black-walled µClear 96-well plates at a density of 4000 cells per well in 100 µL of complete growth media and allowed to adhere for 24 h at 37 °C with 5% CO₂. The media was removed and replaced with complete media containing 5 µM, 10 µM, 50 µM, or 100 µM artesunate, 0.1% DMSO as a negative control, or 25 μM cisplatin as a positive control. Cells were incubated with drugs for 48 h and then fixed for 15 min at room temperature in 4% paraformaldehyde. We used 0.25% Triton X-100 to permeabilize the cells for 15 min and then blocked in 0.1% bovine serum albumin (BSA) for one hour. We assessed DNA damage with immunofluorescent staining for phosphorylated histone H2AX (pH2AX) using the HCS DNA Damage Kit (Invitrogen). We used the Cell-Insight CX7 High Content Analysis Platform for imaging and HCS Studio software to quantify the nuclear pH2AX signal (both ThermoScientific). We completed the statistical analysis of pH2AX signal on GraphPad Prism (version 5.01).

### 2.5. Flow Cytometric Analysis of Cell Cycle

We propagated UWB1 and Caov-3 cells in culture flasks under standard cell culture conditions outlined above. For the staining solution, 2 mg DNase-free Rnase A (Sigma) and 200 μL of 1 mg/mL propidium iodide were added to 10 mL of 0.1% (*v*/*v*) Triton X-100 in PBS. Treatment media was made by serially diluting stock 200 μM artesunate solutions into cell line-specific media on the day of treatment. Once cells cultures reached confluence, we removed the growth media from the treatment flask and added the artesunate treatment media at a concentration of 10 μM. Cells were collected after 24 or 48 h, washed in PBS, and resuspended in 0.5 mL PBS before transferring to tubes containing 4.5 mL 70% EtOH for fixation. Cells were fixed at least overnight at −20°C. Fixed cells were washed in PBS and resuspended in 1 mL of staining solution. We sorted the propidium iodide stained cells with the LSR II cell analyzer. We performed the analysis with ModFit LT v3.3 software. Statistical analysis comparing the percentage of cells in each phase of the cell cycle (G1, S, or G2) was performed using GraphPad Prism (version 5.01).

### 2.6. Drug Administration Sequence Assay

In a similar fashion to the protocols mentioned above in Section 2.2, we plated cells in a standard 96-well plate at a cell density of 3000 cells/100 μL and incubated for 24 h. We diluted artesunate, carboplatin, and paclitaxel stock solutions with DMSO and media to achieve a final concentration of 40 μM, 16 μM, and 32 μM, respectively. Each drug was added, as indicated, for 24 h and treatment media were then replaced with fresh media. Drug administration sequences were artesunate on day 1 (D1A) or day 2 (D2A), carboplatin and paclitaxel on day 2 (D2C/T), carboplatin, paclitaxel, and artesunate on day 2 (D2C/T/A), or artesunate on day 1 followed by carboplatin and paclitaxel on day 2 (D1A; D2C/T). Cells were incubated at standard growth conditions for a total of 72 h. Viability measurements were determined using the CellTiter-Glo 2.0 viability assay (Promega). Luminescence was measured using a Varioskan LUX multimode microplate reader (ThermoFisher Scientific). Statistical analysis was performed using GraphPad Prism (v5.01).

## 3. Results

### 3.1. Artesunate Has Antineoplastic Activity in Ovarian Cancer

To evaluate the antineoplastic activity of artesunate in ovarian cancer, we assessed the dose-dependent effect of artesunate on the viability of three epithelial ovarian cancer cell lines: Caov-3 (adenocarcinoma), UWB1.289 (high-grade serous carcinoma with a BRCA1 mutation), and OVCAR-3 (adenocarcinoma) using the CellTiter-Glo 2.0 assay. The IC50 of artesunate in all three cell lines was in the low to mid micromolar range (Figure 1); specifically, the IC50 was 26.91 µM (95% confidence interval 6.287–115.2 µM) in UWB1, 15.17 µM (10.49–21.93 µM) in Caov-3, and 4.67 µM (3.280–6.638 µM) in OVCAR-3 cells. A One-way ANOVA analysis showed no significant difference (*p* > 0.05) in the IC50 between these three cell lines. The IC50 detected for all three ovarian cancer cell lines tested is consistent with previously established findings [15] and within range of therapeutically achievable in vivo plasma concentrations (approximately 20 μM) [14,21].

In addition to assessing the sensitivity of established ovarian cancer cell lines to artesunate, we also treated a panel of six primary serous ovarian cancer organoids with increasing doses of artesunate and assessed cell viability using an MTS assay. Two of the organoid lines, 1226 and 1254, had relatively high IC50 values of 4.478 µM (367.3 nM–54.48 µM) and 2.72 µM (1.564–4.757 µM), respectively (Figure 1c). The other 4 lines were more sensitive to artesunate with IC50 values in the nanomolar range as follows: 2238 = 89.53 nM (40.63–197.3 nM), 2326 = 62.55 nM (5.359–730.1 nM), 1236 = 17.42 nM (1.345–225.5 nM), and 1267= 9.22 nM (0.8596–98.94 nM). The IC50 values calculated for all six serous ovarian cancer organoid models fall within the range of therapeutically achievable plasma concentrations.

### 3.2. RNAseq Analysis Comparing Resistant and Sensitive Organoids

RNAseq analysis was performed on untreated primary organoid ovarian cancer cell lines. After stratifying the cell lines into sensitive (2238, 2326, 1236, 1267) or resistant (1226 and 1254) based on their IC50 values, analysis revealed 1042 genes differentially expressed between the two groups of organoid models (Appendix A). A heatmap was generated for the top 26 most significantly differentiated genes between these two groups (Figure 2). These genes were further interrogated in the Edge software package to reveal 588 pathways significantly upregulated in the resistant organoid models versus 211 pathways that were downregulated. Table 1 lists a subset of 38 pathways of interest for artesunate use in cancer and indicates which of the differentially expressed genes are associated with each pathway [8,11,12,15,20,27]. These 38 pathways of interest contain 8 genes with increased expression and 2 genes with decreased expression in the resistant organoid group. All of the 10 genes identified in these pathways are among the most significantly differentially expressed genes seen in Figure 2. The false discovery rate (FDR) was used to adjust for multiple comparisons in the RNAseq analysis. A multiple comparison correction was not performed on the *p*-values of the identified differentially expressed pathways. In the artesunate-resistant organoids, pathways involved in the cellular response to oxidative stress were upregulated, while pathways that negatively regulate oxidative phosphorylation and mitochondrial function were downregulated. Additionally, pathways related to cell cycle progression, specifically G1/S transition, were upregulated in the ovarian organoid models more resistant to artesunate.

### 3.3. Clinically Relevant Concentrations of Artesunate Induce G1 Arrest, but Not DNA Damage

After showing that artesunate can reduce cell viability in ovarian cancer cells, we explored potential resistance mechanisms using the pathway analysis from RNAseq comparing artesunate-sensitive and -resistant organoid models. Previously published findings from multiple cancer types have implicated several mechanisms of action of artesunate, including increased reactive oxygen species (ROS) that can induce DNA damage and cell cycle arrest [15,17,28]. Caov-3 cells are used because of their close correlation to IC50s previously published as well as their known platinum sensitivity which correlates to most high-grade serous ovarian cancers [15,29]. To assess the ability of artesunate to induce DNA damage in Caov-3 cells, we quantified the immunofluorescent staining for pH2AX, a marker of double-strand breaks, following a 48 hr treatment with concentrations of artesunate ranging from 5–100 μM and 25 μM cisplatin as a positive control (Figure 3). Cells treated with 0.1% DMSO as a control had a mean pH2AX signal of 443.2 +/− 76.38, while treatment with 25 μM cisplatin resulted in a mean signal of 2517 +/− 230.4. The only artesunate treatment that significantly increased DNA damage was 100 μM artesunate with a mean pH2AX signal of 617.0 +/− 31.96. Treatment with 5, 10, or 50 μM resulted in pH2AX measurements of 382.8 +/− 39.58, 370.2 +/− 9.283, and 393.8 +/− 58.79, respectively. Only the highest concentration of artesunate (100 μM) and cisplatin resulted in a significant increase in DNA damage compared to vehicle-treated control cells as assessed by a one-tailed unpaired *t*-test (*p* = 0.0486 for 100 μM artesunate and *p* = 0.0034 for 25 μM cisplatin). Therefore, while artesunate was able to induce DNA damage, as previously reported [4], we did not observe this effect at clinically relevant concentrations in Caov-3 cells and did not pursue this further in additional cell lines.

The second mechanism of action for artesunate we investigated was the induction of cell cycle arrest. We assessed cell cycle progression by propidium iodide staining and flow cytometry of Caov-3 and UWB1 cells following treatment with 10 μM artesunate or 0.1% DMSO (vehicle control) for 24 and 48 h. Representative histogram images of the propidium iodide staining and overlaid cell cycle analysis are in Appendix A (Caov-3) and Appendix A (UWB-1). The percentages of cells in G1 and S phase with and without 10 μM artesunate treatment were determined from three independent experiments. In Caov-3 cells, 48-h artesunate treatment resulted in an increased percentage of cells in G1 (78.98% +/− 0.9546 compared to 61.23% +/− 1.789 in vehicle-treated cells). UWB1 cells had 65.35% +/− 0.0849 cells in G1 after artesunate treatment compared to 60.35% +/− 2.418 of control cells (Figure 4a,b). Both cell lines showed a significant increase in cells in G1 (*p* = 0.0032 in Caov-3 and *p* = 0.0499 in UWB1; one-tailed unpaired *t*-test). In a further subgroup analysis, the increased percentage of Caov-3 and UWB1 cells in the G1 phase following treatment with artesunate was accompanied by a significant decrease in cells in the S phase (*p* = 0.0009 and *p* = 0.0497, respectively). In Caov-3 cells, 48-h artesunate treatment resulted in a decreased percentage of cells in S phase (12.14% +/− 0.1556 compared to 24.42% +/− 0.7354 in vehicle-treated cells). UWB1 cells had 18.87% +/− 2.779 cells in S phase after artesunate treatment compared to 26.25% +/− 2.234 of control cells (Figure 4c,d). These experiments demonstrate that in the cell lines tested, clinically relevant concentrations of artesunate induce G1 arrest but not DNA damage.

### 3.4. Addition of Artesunate to Carboplatin and Paclitaxel Improves Antineoplastic Activity

The current primary treatment regimen in ovarian cancer is a platinum/taxane doublet, such as carboplatin and paclitaxel. The favorable side effect profile and its effects on cell viability make artesunate a possible addition to the current standard of care regimen. Therefore, we next assessed the effect of adding artesunate to cells treated with both paclitaxel and carboplatin. Since artesunate induces cell cycle arrest in the G1 phase, we evaluated artesunate as either a pretreatment (24 h prior) or concurrently with carboplatin and paclitaxel. In both cell lines, treatment with artesunate alone on either day 1 or day 2 resulted in a decreased cell viability to 54.60–63.43% compared to vehicle-treated control cells (Figure 5). In Caov-3 cells, treatment with carboplatin/paclitaxel resulted in reduced cell viability to 29.60% +/− 7.780, which was significantly decreased when artesunate was added concurrently (11.55% +/− 5.917, *p* < 0.05 One-way ANOVA) but not when cells were pretreated with artesunate (33.16% +/− 3.349, *p* > 0.005). Similarly, in UWB1, cell viability was decreased to 62.06% +/− 9.389 when treated with carboplatin and paclitaxel, which was further decreased to 39.65% +/− 6.850 (*p* < 0.05, One-way ANOVA) with the concurrent addition of 40 μM artesunate. In these cells, pretreatment with artesunate resulted in cell viability of 56.41% +/− 2.148, which was not statistically significant when compared to cells treated with carboplatin and paclitaxel. Given that carboplatin works in the S phase and paclitaxel in the M phase, it is not surprising that pretreatment with artesunate, which causes G1 arrest, did not improve the regimen’s effectiveness while concurrent administration did. Future clinical studies should consider concurrent administration of the three medications.

## 4. Discussion

The discovery of artemisinin as an anti-malarial compound in 1967 was a significant medical advance, with Tu Youyou being awarded the Nobel prize in 2015 for this same discovery [11]. Our understanding of its physical and biochemical properties over the last half-century has led to the development of artesunate as a successful malaria treatment. This safe and well-tolerated medicine has also shown promising antineoplastic activity across several cancer types. Efferth and colleagues demonstrated in vitro activity across a panel of 50 established cancer cell lines while also defining several of the molecular modes of action [12]. In this study, which is consistent with other published reports [9,15,20,21,27,28,30,31], we demonstrate that artesunate inhibits cell viability with IC50 values in the micromolar range across a panel of ovarian cancer cell lines. Additionally, several patient-derived 3D tumor organoids, a more accurate model of in vivo tumor biology with the potential to correlate in vivo *and* in vitro chemotherapy sensitivities, were even more sensitive to artesunate with IC50 values ranging from 9.22 nM–4.478 μM (Figure 1c) [32]. These concentrations are achievable clinically, with one study of adults with severe malaria receiving a 2.4 mg/kg dose of artesunate twice a day demonstraiting a Cmax of approximately 3200 ng/mL (8.325 μM) [10], which is also consistent with concentrations found in a phase 1 trial of artesunate in solid tumor malignancies [19]. Encouraged by the cytotoxicity demonstrated at clinically achievable doses, we evaluated potential mechanisms of action of artesunate in ovarian cancer.

Efferth and colleagues demonstrated that CDC25A protein, which governs G1 cell entry into S1, is downregulated with artesunate treatment [12]. Our data showing G1 phase arrest supports these findings (Figure 4). Furthermore, the Greenshield lab determined that cell cycle arrest is dose-dependent with G1 phase arrest at concentrations comparable to our in vitro data at 10 μM [15]. They also showed G2/M phase arrest at 100 μM [15]. Several trends in differential expression of mRNA in artesunate-resistant organoids support these previously proposed mechanisms and are consistent with genes identified across a panel of 55 different tumor types that correlate with artesunate resistance [12]. Pathways involved in the cellular response to oxidative stress and G1/S transition are upregulated, while pathways that negatively regulate oxidative phosphorylation and mitochondrial function are downregulated (Table 1). Although our lab did not show DNA damage at therapeutic concentrations of artesunate, other labs have shown it downregulates RAD51 and inhibits the mTOR pathway [15,21], further sensitizing cells to ROS-induced double-strand break. While we demonstrate DNA damage at 100 μM concentrations, this was not observed at lower concentrations. It remains unclear whether artesunate induces ROS at therapeutic concentrations but genes involved in response to oxidative stress are upregulated in resistant organoid models (Table 1). Other studies have demonstrated that artesunate’s ability to cause oxidative damage is iron-dependent, due to the endoperoxide bridge in artesunate interacting with Heme iron-producing cytotoxic radicles [16] This may explain the lack of ROS at physiologic concentrations observed in iron-poor in vitro systems. Although artesunate is not affected by most common chemotherapy resistance pathways [12,32], EGFR expression did correlate with resistance and EGFR expression was upregulated in the resistant primary organoid models in this study (Figure 2 and Table 1). EGFR is known to induce expression of Bcl-2 and down-regulates BAX, resulting in inhibition of apoptosis. Alternatively, artesunate is reported to inhibit phosphorylation of BAD, promoting the formation of BAD/Bcl-xL complex, which triggers the intrinsic apoptotic cascade involving cytochrome c and may serve as a mechanism for ROS formation-independent cell death [33]. Therefore, it is reasonable to hypothesize that EGFR overexpression in artesunate-resistant ovarian cancer cells could impair the ability of artesunate to induce apoptosis through the BAD/Bcl-xL complex. Additional studies are needed to verify the role of EGFR overexpression in artesunate resistance and to determine if EGFR inhibitors could sensitize ovarian cancer models, which would otherwise harbor innate resistance, to artesunate treatment.

While most of the artesunate safety data are taken from malaria treatment studies with a short duration of administration, a small study in patients with metastatic breast cancer showed good safety and tolerability to oral artesunate dosages of up to 200 mg/day over several years [34]. Anemia and diarrhea are the most common adverse effects. [32,35]. Currently, there are several clinical trials investigating artesunate’s anticancer effects, but most are early phase [17]. There are no published reports of artesunate outcomes in ovarian cancer, but one patient with metastatic ovarian cancer was enrolled in an early-phase trial and had stable disease while receiving eight months of the lowest dose (8 mg/kg) [19]. Although trial designs have varied routes of administration and schedules, intravenous artesunate treatment has been shown to have an maximum tolerated dose of 18 mg/kg on a D1/D8, 3 week administration [19] that could be incorporated into standard dosing schedules of frontline carboplatin and paclitaxel.

The administration schedule of chemotherapies is vitally important when utilizing multi-drug regimens as many drugs have mechanisms of action that are dependent on the cell cycle. Taxol has sequence-dependent cytotoxicity. In ovarian cancer, sequential administration of paclitaxel followed by carboplatin is the standard of care. If carboplatin is administered first, there is an antagonistic interaction with paclitaxel that can be observed along with increased toxicity [35]. Taxol has been reported to have similar sequence-dependence with etoposide, 5-FU, and several other agents [7]. Furthermore, since we observed G1 phase arrest with artesunate administration, it was necessary to learn more about the potential interactions with carboplatin and paclitaxel, which is M phase-dependent. As anticipated, we saw a decrease in cell viability with coadministration versus pretreatment with artesunate. This finding should help guide further investigative efforts in incorporating artesunate into ovarian cancer treatment.

To our knowledge, this is the first paper to describe the increased anticancer effect of artesunate in combination with carboplatin and paclitaxel in ovarian cancer. Strengths of this work include the large number of ovarian cell lines and organoids that show the consistent anticancer activity of artesunate, the demonstration of the added activity of artesunate in combination with standard therapies, and our RNAseq studies that support the role of cell cycle arrest and generation of ROS as the primary mechanism of artesunate activity.

Artesunate has been shown to have single-agent activity in a variety of cancer types [19,34]. The observed activity in ovarian cancer as a single agent in in vitro studies [15] along with its favorable toxicity profile [10,36] make it a promising candidate to study in ovarian cancer treatment. Additionally, our research supports further evaluation of artesunate combined with paclitaxel and carboplatin in animal models and future clinical trials.

## Figures and Tables

**Figure 1 diagnostics-11-00395-f001:**
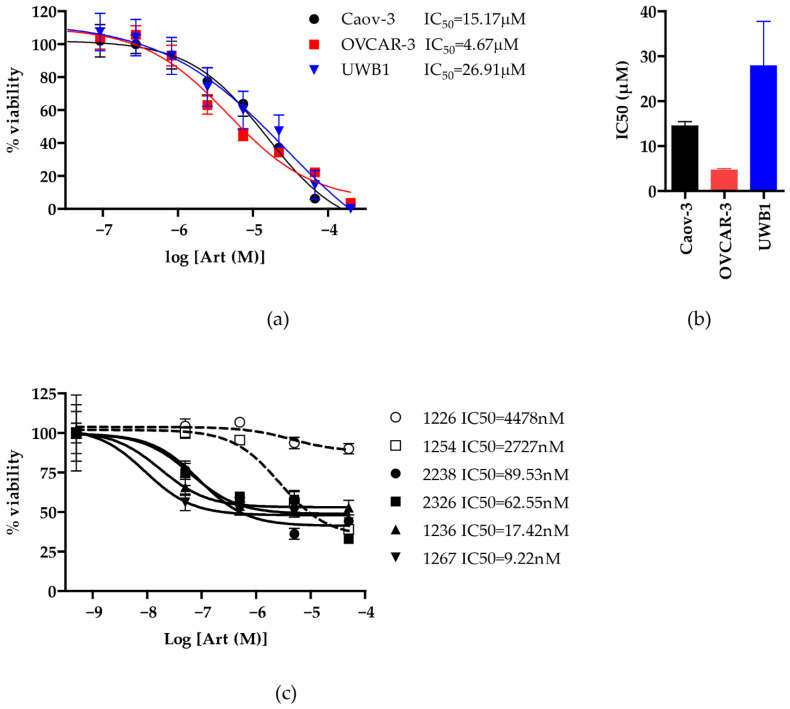
Artesunate sensitivities across 3 commercially available ovarian cancer cell lines and 6 novel 3D ovarian cancer organoids. (**a**) Human ovarian cancer cell lines Caov-3, OVCAR-3, and UWB1.289 cells were treated with serially diluted concentrations of artesunate for 72 h. CellTiter-Glo 2.0 viability assay (Promega) was used to calculate percent viability from the proliferation of treated versus vehicle-treated control cells. The mean +/− SD from three independent experiments is shown graphically and IC50s were calculated using a variable slope non-linear regression line. (**b**) The mean IC50 of artesunate treatments in each cell line is graphed with SD. Utilizing a one-way ANOVA with Bonferroni’ Multiple Comparison Test, all IC50s were not significantly different with *p* > 0.05. (**c**) 3D ovarian organoids were established from 6 patients and were treated similarly with serially diluted concentrations of artesunate for 72 h. Viability was measured with MTS assay (Promega) and the signal for each drug treatment was normalized to a matched vehicle-treated control. The mean % viability +/− SD is shown graphically and IC50s were calculated using a variable slope non-linear regression line.

**Figure 2 diagnostics-11-00395-f002:**
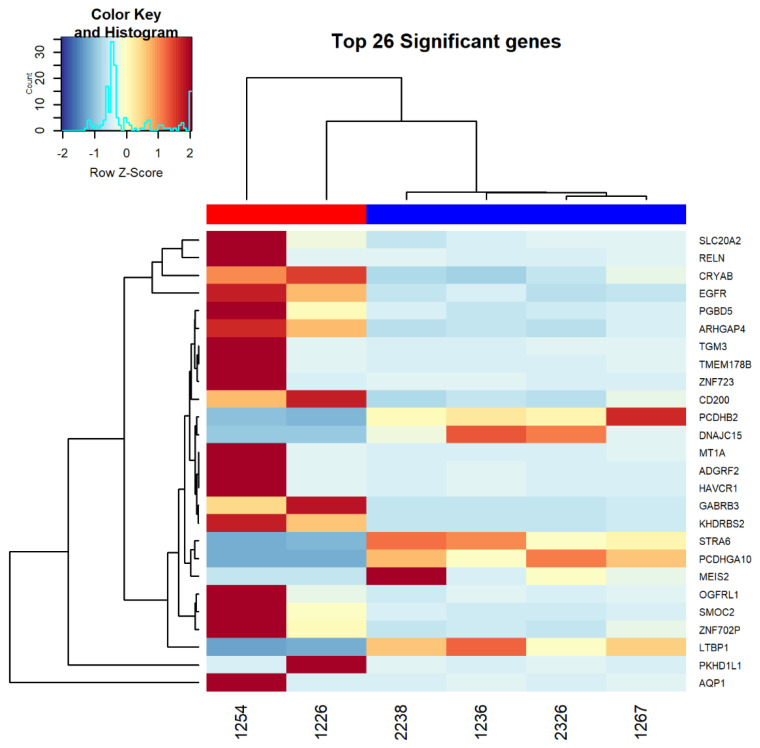
Heat map showing the top 26 most significantly differentially expressed genes between ovarian organoid models that are resistant to artesunate (1254 and 1226) compared to models that are sensitive to artesunate (2238, 2326, 1267, and 1236). The red bar along the top of the heat map denotes the resistant models while the blue bar indicates the sensitive models. RNAseq analysis was performed on RNA extracted from untreated cells.

**Figure 3 diagnostics-11-00395-f003:**
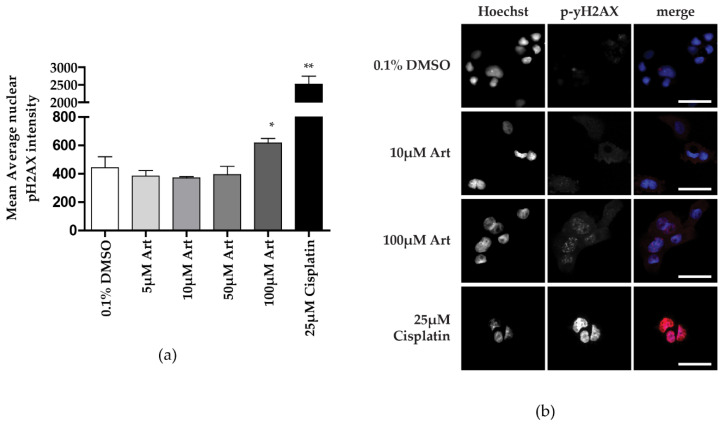
DNA damage assay as average nuclear pH2AX intensity. (**a**) The average nuclear intensity of pH2AX staining was quantified in Caov-3 cells treated for 48 h with artesunate concentrations ranging from 5–100 μM with 25 μM cisplatin treatment as a positive control. The mean signal +/− SD was graphed and a one-tailed *t*-test was performed (* *p* = 0.0486, ** *p* = 0.0034) (**b**) Representative images of cells treated with 0.1% DMSO (vehicle), 10 μM artesunate, 100 μM artesunate, or 25 μM cisplatin (positive control for DNA damage).

**Figure 4 diagnostics-11-00395-f004:**
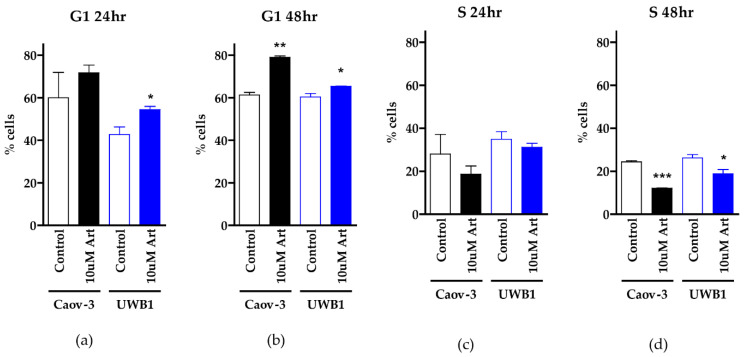
Cell cycle analysis using propidium iodide staining. The percentage of Caov-3 and UWB1 cells in G1 after 24 h (**a**) or 48 h (**b**) treatment with 0.1% DMSO (control) or 10 μM artesunate is graphed as the mean +/− SD from three independent experiments. A one-tailed unpaired *t*-test revealed a statistically significant increase in cells in G1 in UWB1 cells treated for either 24 or 48 h (* *p* < 0.05) and in Caov-3 cells treated with artesunate for 48 h (** *p* < 0.01). The concurrent analysis of the percentage of cells in S phase following 24 (**c**) or 48 (**d**) hour treatment with artesunate reveals a significant decrease in cells in S phase after 48 h (* *p* < 0.05; *** *p* < 0.001).

**Figure 5 diagnostics-11-00395-f005:**
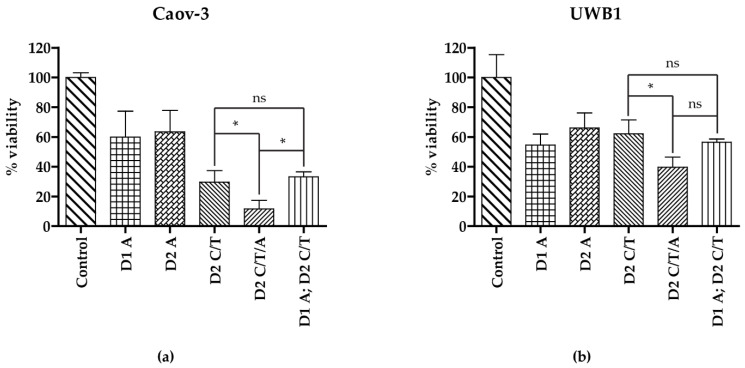
Drug Administration Sequence Assay for artesunate, carboplatin, and paclitaxel. Cells were treated with artesunate on day 1 (D1A) or day 2 (D2A), carboplatin and paclitaxel on day 2 (D2C/T), carboplatin, paclitaxel, and artesunate on day 2 (D2C/T/A), or artesunate on day 1 followed by carboplatin and paclitaxel on day 2 (D1A; D2C/T). The 24 h treatment concentrations for artesunate, carboplatin, and paclitaxel were 40 μM, 16 μM, and 32 μM, respectively. Percent viability was calculated utilizing the CellTiter-Glo 2.0 viability assay following treatment with carboplatin/paclitaxel and/or artesunate compared to DMSO-treated (control) cells, as indicated, shown graphically as the mean +/− SD in Caov-3 (**a**) and UWB1 (**b**) cells. Statistical differences were assessed by One-way ANOVA (ns—not significant and * *p* < 0.05). In both cell lines, the addition of artesunate as a concurrent treatment with carboplatin/paclitaxel resulted in a significant decrease in viable cells.

**Table 1 diagnostics-11-00395-t001:** Pathway analysis of genes differentially expressed between the resistant (1226 and 1254) and sensitive (2238, 2326, 1236, 1267) organoid models. Gene ontology IDs are listed under pathway ID.

General Category	Pathway ID	Pathway Description	*p*-Value	Differentially Expressed Genes
ROS/oxidative phosphorylation	GO:0000302	response to reactive oxygen species	↑ 0.0008	↑ AQP1, ↑ CRYAB, ↑ EGFR
	GO:0006979	response to oxidative stress	↑ 0.0056	↑ AQP1, ↑ CRYAB, ↑ EGFR
	GO:0034614	cellular response to reactive oxygen species	↑ 0.0077	↑ AQP1, ↑ EGFR
	GO:2000377	regulation of reactive oxygen species metabolic process	↑ 0.0098	↑ CRYAB, ↑ EGFR
	GO:0072593	reactive oxygen species metabolic process	↑ 0.0212	↑ CRYAB, ↑ EGFR
	GO:0034599	cellular response to oxidative stress	↑ 0.0241	↑ AQP1, ↑ EGFR
	GO:2000378	negative regulation of reactive oxygen species metabolic process	↑ 0.045	↑ CRYAB
	GO:0090324	negative regulation of oxidative phosphorylation	↓ 0.0018	↓ DNAJC15
	GO:0002082	regulation of oxidative phosphorylation	↓ 0.0074	↓ DNAJC15
	GO:0006119	oxidative phosphorylation	↓ 0.0352	↓ DNAJC15
mitochondrial functon/cellular respiration	GO:1902957	negative regulation of mitochondrial electron transport, NADH to ubiquinone	↓ 0.0005	↓ DNAJC15
	GO:1905447	negative regulation of mitochondrial ATP synthesis coupled electron transport	↓ 0.0005	↓ DNAJC15
	GO:1901856	negative regulation of cellular respiration	↓ 0.0010	↓ DNAJC15
	GO:1902956	regulation of mitochondrial electron transport, NADH to ubiquinone	↓ 0.0013	↓ DNAJC15
	GO:1905446	regulation of mitochondrial ATP synthesis coupled electron transport	↓ 0.0018	↓ DNAJC15
	GO:0005744	TIM23 mitochondrial import inner membrane translocase complex	↓ 0.0036	↓ DNAJC15
	GO:0043457	regulation of cellular respiration	↓ 0.0076	↓ DNAJC15
	GO:0006120	mitochondrial electron transport, NADH to ubiquinone	↓ 0.0135	↓ DNAJC15
	GO:0042775	mitochondrial ATP synthesis coupled electron transport	↓ 0.0235	↓ DNAJC15
	GO:0042773	ATP synthesis coupled electron transport	↓ 0.0238	↓ DNAJC15
	GO:0006626	protein targeting to mitochondrion	↓ 0.0248	↓ DNAJC15
	GO:1990542	mitochondrial transmembrane transport	↓ 0.0255	↓ DNAJC15
	GO:0022904	respiratory electron transport chain	↓ 0.0285	↓ DNAJC15
	GO:0098800	inner mitochondrial membrane protein complex	↓ 0.0332	↓ DNAJC15
	GO:0072655	establishment of protein localization to mitochondrion	↓ 0.0345	↓ DNAJC15
	GO:0045333	cellular respiration	↓ 0.0473	↓ DNAJC15
cell cycle progression	GO:1900087	positive regulation of G1/S transition of mitotic cell cycle	↑ 0.0302	↑ EGFR
	GO:0045740	positive regulation of DNA replication	↑ 0.0302	↑ EGFR
	GO:1902808	positive regulation of cell cycle G1/S phase transition	↑ 0.0384	↑ EGFR
	GO:0045787	positive regulation of cell cycle	↑ 0.0391	↑ SMOC2, ↑ EGFR, ↓ MEIS2
response to metal ion	GO:0010038	response to metal ion	↑ 0.0034	↑ AQP1, ↑ MT1A, ↑ EGFR
	GO:0071248	cellular response to metal ion	↑ 0.0005	↑ AQP1, ↑ MT1A, ↑ EGFR
Cell death/apoptosis	GO:0060548	negative regulation of cell death	↑ 0.0011	↑ AQP1, ↑ GABRB3, ↑ CD200,↑ CRYAB, ↑ EGFR
	GO:0043067	regulation of programmed cell death	↑ 0.0375	↑ AQP1, ↑ GABRB3, ↑ CRYAB,↑ EGFR
ATP metabolic process	GO:1903579	negative regulation of ATP metabolic process	↓ 0.0064	↓ DNAJC15
	GO:1903578	regulation of ATP metabolic process	↓ 0.0300	↓ DNAJC15
response to antineoplastic agent	GO:0097327	response to antineoplastic agent	↑ 0.0029	↑ AQP1, ↑ EGFR
phosphatidylinositol-mediated signaling	GO:0048015	phosphatidylinositol-mediated signaling	↑ 0.0097	↑ RELN, ↑ EGFR

↑: indicates upregulation of pathway or increased gene expression in artesunate-resistant organoid models; ↓: indicates downregulation of pathway or decreased gene expression in artesunate-resistant organoid models.

## Data Availability

The data presented in this study are available on request from the corresponding author. The data are not publicly available due to HIPPA.

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
