# Peer review of "Preclinical Evaluation of Artesunate as an Antineoplastic Agent in Ovarian Cancer Treatment"

_diagnostics, 2021, doi:10.3390/diagnostics11030395_

Round 1
Reviewer 1 Report
The authors have systematically investigated the role of antineoplastic agent on the treatment of ovarian cancer. This research is very urgent and the data will be interested to the scientists and doctors in this domain. Personally, this submission can be accepted by Diagnostics without change.
Author Response
Thank you for your review. Several changes have been made to the paper at the request of additional editors. We hope you find these changes acceptable and look forward to publication of this paper.
Reviewer 2 Report
The manuscript by McDowell, et al describes the potential as Aresunate as a treatment for ovarian cancer. While this is an interesting field of study, there are significant concerns.
The first major concern is the poor/lacking descriptions of experimental design, details, and analysis. Most experiments are not sufficiently described in the results. Additionally, the figure legends are not detailed enough to interpret the data. This is a major flaw. All results and legends need to add significantly more details. For example, for the RNA-seq, describe the experimental set up and controls for the cell lines used. It is not clear what is being done for the experiments in figure 4. The legends do not describe the abbreviations on the x axis. Experimental details are lacking in 3.4 (both results and figure legends).
Why were only CAOV3 cell used in Figure 2? The legend for figure 2 does not say what cells are being used. All legends should state cell lines being used.
Flow cytometry histograms with controls for figure 3 should be shown.
Another major concern is the temptation to overstate results or a lack of specificity in terms. Can the terms dose timing and antineoplastic activity be better described and more specific language should be used instead? In 3.1 the text states that antineoplastic activity is being measured. This is an overstatement. 3.1 only demonstrates the IC50 values more experimentation would be needed to conclude a true antineoplastic effect. What assay was used to determine IC50 in figure 1? This is not stated in the results section or legend.
In the methods section it says that the Celltiter 2.0 glow assay measures proliferation. The assay is a measure of cell viability. It cannot ascertain if changes in cell viability of a cell population are due to proliferation, cell death, differentiation, senescence, etc. Please later conclusions.
Finally, the mechanism of action is weak. None of the genes identified via RNA-seq are provided or validated in the different cell lines (No RNA-seq data is reported, only the pathways. That seems like it should be required data). Providing data on genes that are altered by Artesunate would enhance the paper. Showing changes in phases of the cell cycle does not significantly demonstrate how Artesunate leads to reduced viability. evidence that Artesunate augments cell cycle machinery would be preferred.
Author Response
Thank you for your review. We have found you requests insightful and have attempted to address them. We have added detail to the methods section for each of the experiments as well as the figures and associated legends which hopefully clarifies things. We have added additional details on the RNA-seq. Specifically, we clarified the cell population that this data was extracted from. These primary ovarian cancer cell lines were NOT treated with artesunate before the RNA-seq was run. This information cannot be used to look at genes altered by artesunate. Instead we have used it to help guide further investigation into pathways that could be implicated in not only mechanism of action but also mechanism of resistance. We saw a significant difference in several pathways that correlate with previously published data as well as the findings in our paper. The specific genes were not listed because only 6 cell lines were used and there is a high degree of variability and error with such a small sample size.
There is a supplemental figure 1 that will be incorporated into the paper that will have the flow cytometry histograms for UWB1 and Caov-3 from a representative experiment.
We agree that overstating results should be avoided and that is one of the reasons for choosing “antineoplastic”. As you have correctly stated, the Celltiter 2.0 glow assay measures cell viability and cannot ascertain if changes in cell viability of a cell population are due to proliferation, cell death, differentiation, senescence, etc. Furthermore, the effects that are seen on the treated cells are dose dependent. DNA damage was not observed until supratherapeutic treatment concentrations are used. Furthermore, we attempted to use concentrations that would mimic in-vivo effect and found that at the 10 uM treatment used in the flow cytometry assay, minimal apoptotic signals were observed in the samples. As mentioned in the discussion, several papers have looked at the mechanism of action with widely varying results. For these reasons, we chose to use the broad term antineoplastic to describe artesunate’s activity.
We have changed our wording on the Dose timing to “Drug Administration Sequence” to better describe the . Figure 2 shows the data from Caov-3 cells which were used because of their close correlation to IC50s previously published as well as their known platinum sensitivity which correlates to most high grade serous ovarian cancers. While artesunate was able to induce DNA damage, we did not observe this effect at clinically relevant concentrations in Caov-3 cells and did not pursue this further in additional cell lines.
While we appreciate the in-depth review, we kindly disagree with the final statement. The mechanism of action has been interrogated by several other labs with varying conclusions about specific mechanism of actions. While we partially agree with these findings, our paper attempted to show that this drug likely has a multimodal mechanism that is dose dependent. The inclusion of the RNA-seq data was to help validate the finding of G1 phase arrest as well as reinforce the role of previously published mechanisms. As mentioned, prior publications have shown downregulation of RAD51 which may be a mechanism of sensitizing cells to ROS despite the lack of DNA damage observed in our study. Furthermore, several pathways for potential resistance were seen that can help guide further investigation. Specifically, our lab is interested looking at the specific genes altered by artesunate but is still being investigated. Additionally, there is a large amount of safety data on this drug’s use in malaria treatment and we wanted to offer a preclinical perspective on how this drug may fit into standard cytotoxic chemotherapy regimen and was a major focus in the conclusions.
Reviewer 3 Report
The work deals with an interesting topic. The cytostatics used in the treatment of ovarian cancer have not changed for many years. The manuscript needs minor corrections. The authors should explain more why they chose this particular anti-malarial drug. Similarly, in the discussion, they should refer to other drugs that have been tried to be added to routine first-line chemotherapy with varying success.
Author Response
Thank you for your review. Several changes have been made to the paper at the request of additional editors and hope you find these changes acceptable. Additional background on previous investigations into drug combinations used for ovarian cancer treatment in the introduction. Specifically, one key study through the GOG, which is the main scientific community for gynecologic oncology, has investigated the role of a third cytotoxic chemotherapy into the standard regimen and found no benefit (GOG 182).
The University of Kentucky has been investigating the growth and cultivation of the artemisia annua plant that is used in the production of artesunate. This crop is planted, grown, harvested, and dried like the tobacco plant. Kentucky has many tobacco growers used to this cultivation process and is one of the reasons this region has been chosen. Specifically, the Department of Pharmacy and the Markey Cancer Center were interested in artesunate because of the preclinical data that has already shown potential activity in cancer treatment and looked to expand on that knowledge. This background information was thought to be out of the scope of this paper.
Round 2
Reviewer 2 Report
See original suggestions.
To me it looks like very little was done to address the original concerns. There are few improved changes to methods, legends, etc.
I find the description of the RNA-seq in the response more confusing. I less see the point of why the RNA-seq was done. Furthermore, they state that they can identify pathways but not genes due to variability. This doesn't sound like reproducible data.
The authors also disagree when I asked for more mechanism. Mechanistic studies may have been performed by other labs, but little is shown in this paper.
My comments to the authors would be to address the original comments.
Author Response
Reviewer 2, comment 1, first review: The first major concern is the poor/lacking descriptions of experimental design, details, and analysis. Most experiments are not sufficiently described in the results. Additionally, the figure legends are not detailed enough to interpret the data. This is a major flaw. All results and legends need to add significantly more details. For example, for the RNA-seq, describe the experimental set up and controls for the cell lines used. It is not clear what is being done for the experiments in figure 4. The legends do not describe the abbreviations on the x axis. Experimental details are lacking in 3.4 (both results and figure legends).
Reviewer 2, comment 1, second review: There are few improved changes to methods, legends, etc
Author response, first review: We have added detail to the methods section for each of the experiments as well as the figures and associated legends.
Author response, second review: No additional changes have been made.
Reviewer 2 comment 2: Why were only CAOV3 cell used in Figure 2? The legend for figure 2 does not say what cells are being used. All legends should state cell lines being used.
Author response, first review: We have included the cell line in each of the figures. Figure 2 shows the data from Caov-3 cells which were used because of their close correlation to IC50s previously published as well as their known platinum sensitivity which correlates to most high grade serous ovarian cancers. While artesunate was able to induce DNA damage, we did not observe this effect at clinically relevant concentrations in Caov-3 cells and did not pursue this further in additional cell lines.
Author response, second review: No additional changes have been made.
Reviewer 2 comment 3: Flow cytometry histograms with controls for figure 3 should be shown.
Author response, first review: Flow cytometry histograms have been added as supplementary figure 2
Author response, second review: No additional changes have been made.
Reviewer 2, comment 4: Another major concern is the temptation to overstate results or a lack of specificity in terms. Can the terms dose timing and antineoplastic activity be better described and more specific language should be used instead? In 3.1 the text states that antineoplastic activity is being measured. This is an overstatement. 3.1 only demonstrates the IC50 values more experimentation would be needed to conclude a true antineoplastic effect. What assay was used to determine IC50 in figure 1? This is not stated in the results section or legend.
Author response, first review: We agree that overstating results should be avoided and that is one of the reasons for choosing “antineoplastic”. As you have correctly stated, the Celltiter 2.0 glow assay measures cell viability and cannot ascertain if changes in cell viability of a cell population are due to proliferation, cell death, differentiation, senescence, etc. Furthermore, the effects that are seen on the treated cells are dose dependent. DNA damage was not observed until supratherapeutic treatment concentrations are used.
Furthermore, we attempted to use concentrations that would mimic in-vivo effect and found that at the 10 uM treatment used in the flow cytometry assay, minimal apoptotic signals were observed in the samples. As mentioned in the discussion, several papers have looked at the mechanism of action with widely varying results. For these reasons, we chose to use the broad term antineoplastic to describe artesunate’s activity.
We have changed our wording on the Dose timing to “Drug Administration Sequence”
Author response, second review: Section 3.1 was revised to change “Antineoplastic” to “Cell Viability” and to include that the Cell titer glow assay was used; Section 2.2 was revised to clearly indicate that cell viability was measured by the cell titer glow assay.
Reviewer 2, comment 5: In the methods section it says that the Celltiter 2.0 glow assay measures proliferation. The assay is a measure of cell viability. It cannot ascertain if changes in cell viability of a cell population are due to proliferation, cell death, differentiation, senescence, etc. Please later conclusions.
Author response, first review: NA
Author response, second review: Line 372. “anticancer” was replace with “cell viability”. See responses above for revisions to section 3.1 and 2.2
Reviewer 2, Comment 6: Finally, the mechanism of action is weak. None of the genes identified via RNA-seq are provided or validated in the different cell lines (No RNA-seq data is reported, only the pathways. That seems like it should be required data). Providing data on genes that are altered by Artesunate would enhance the paper. Showing changes in phases of the cell cycle does not significantly demonstrate how Artesunate leads to reduced viability. evidence that Artesunate augments cell cycle machinery would be preferred.
Author response, first review: While we appreciate the in-depth review, we kindly disagree with the final statement. The mechanism of action has been interrogated by several other labs with varying conclusions about specific mechanism of actions. While we partially agree with these findings, our paper attempted to show that this drug likely has a multimodal mechanism that is dose dependent. The inclusion of the RNA-seq data was to help validate the finding of G1 phase arrest as well as reinforce the role of previously published mechanisms. As mentioned, prior publications have shown downregulation of RAD51 which may be a mechanism of sensitizing cells to ROS despite the lack of DNA damage observed in our study. Furthermore, several pathways for potential resistance were seen that can help guide further investigation. Specifically, our lab is interested looking at the specific genes altered by artesunate but is still being investigated. Additionally, there is a large amount of safety data on this drug’s use in malaria treatment and we wanted to offer a preclinical perspective on how this drug may fit into standard cytotoxic chemotherapy regimen and was a major focus in the conclusions.
Author response, second review: We have added an additional analysis of the RNA Seq data at the gene level and identify EGFR and other candidate genes as overexpressed in inherently resistant cell lines and suggest this as a novel and clinically targetable future direction. Importantly, the primary purpose of our paper was to determine the optimal timing of artesunate for future clinical trials. There is a well-known drug interaction between taxanes and cisplatin, when cisplatin preceeds paclitaxel, antagonism secondary to a G2 arrest created by cisplatin treatment occurs, since taxanes are M phase specific. We both show an antagonistic interaction when artesunate preceeds taxanes, and that this finding is likely mediated by G-phase arrest. While we agree the mechanisms investigated are already known, understanding why the sequence of artesunate and taxanes matters is highly relevent for the future development of this combination.
Reviewer 2, Second review: I find the description of the RNA-seq in the response more confusing. I less see the point of why the RNA-seq was done. Furthermore, they state that they can identify pathways but not genes due to variability. This doesn't sound like reproducible data.
Author response, second review: Section 3.2 was revised to clearly explain that RNA seq was undertaken to understand the mechanisms of inherant artesunate resistance. We have included heat-map of upregulated genes (Figure 1, added the differentially expressed genes to Table 1 and included a supplemental table of all results.